# miRNAs in Uremic Cardiomyopathy: A Comprehensive Review

**DOI:** 10.3390/ijms24065425

**Published:** 2023-03-12

**Authors:** Mario D’Agostino, Davide Mauro, Mariateresa Zicarelli, Nazareno Carullo, Marta Greco, Michele Andreucci, Giuseppe Coppolino, Davide Bolignano

**Affiliations:** Nephrology and Dialysis Unit, Department of Medical and Surgical Sciences, University “Magna-Graecia” of Catanzaro, Viale Europa SNC, 88100 Catanzaro, Italy

**Keywords:** miRNA, uremic cardiomyopathy, ventricular hypertrophy, cardiac fibrosis, chronic kidney disease, end-stage kidney disease

## Abstract

Uremic Cardiomyopathy (UCM) is an irreversible cardiovascular complication that is highly pervasive among chronic kidney disease (CKD) patients, particularly in End-Stage Kidney Disease (ESKD) individuals undergoing chronic dialysis. Features of UCM are an abnormal myocardial fibrosis, an asymmetric ventricular hypertrophy with subsequent diastolic dysfunction and a complex and multifactorial pathogenesis where underlying biological mechanisms remain partly undefined. In this paper, we reviewed the key evidence available on the biological and clinical significance of micro-RNAs (miRNAs) in UCM. miRNAs are short, noncoding RNA molecules with regulatory functions that play a pivotal role in myriad basic cellular processes, such as cell growth and differentiation. Deranged miRNAs expression has already been observed in various diseases, and their capacity to modulate cardiac remodeling and fibrosis under either physiological or pathological conditions is well acknowledged. In the context of UCM, robust experimental evidence confirms a close involvement of some miRNAs in the key pathways that are known to trigger or worsen ventricular hypertrophy or fibrosis. Moreover, very preliminary findings may set the stage for therapeutic interventions targeting specific miRNAs for ameliorating heart damage. Finally, scant but promising clinical evidence may suggest a potential future application of circulating miRNAs as diagnostic or prognostic biomarkers for improving risk stratification in UCM as well.

## 1. Introduction

Chronic Kidney Disease (CKD) is a systemic pathological condition related to a progressive loss in renal function that may affect virtually all organs, particularly the cardiovascular (CV) system. CKD may trigger or worsen existing CV disease (CVD), including atherosclerotic lesions, vascular and cardiac valve calcifications and myocardial fibrosis. As a result, CKD portends an exceedingly higher risk of CV mortality as compared with the general population and this holds true even in the earlier stages of the disease [1]. Roughly 50% of individuals with advanced CKD have CVD which, in turn, accounts for about 40–50% of all deaths in CKD. It is therefore not surprising that the risk of death in CKD patients often exceeds that of progressing to End-Stage Kidney Disease (ESKD) [2], a condition also known as uremia that requires chronic renal replacement support by dialysis or a kidney graft [3].

Various factors contribute to the development of CVD in CKD; these include toxins accumulation, a deranged balance in circulating hormones, enzymes and cytokines, neurovegetative and hemodynamic alterations, oxidative stress, systemic inflammation and sustained anemia. On top of that, traditional CV risk factors, such as hypertension, insulin resistance/diabetes, dyslipidemia and lifestyle habits, such as smoking, may establish a vicious circle, eventually leading to irreversible CV alterations, with a subsequent increase in CV risk [4,5].

Uremic Cardiomyopathy (UCM) is a typical CV complication in which the prevalence in CKD increases in parallel with the severity of the decline in renal function, being the highest in individuals with ESKD on chronic renal replacement therapy. UCM is characterized by abnormal myocardial fibrosis and asymmetric ventricular hypertrophy with subsequent cardiac dysfunction, which is predominantly diastolic [6].

The first triggers of UCM are left ventricular (LV) pressure and volume overload in the setting of the overall uremic state. LV pressure overload promotes hypertrophy by increasing LV wall thickness while volume overload drives an enlargement in chamber size. Although LV hypertrophy (LVH) firstly appears as a compensatory response, a chronic sustained LV overload leads to cardiomyocyte death with consequent systolic heart failure and activation of inflammatory and pro-fibrotic interstitial pathways with ensuing diffuse fibrosis, eventually leading to diastolic dysfunction [6].

Despite various factors having been called into question in the pathogenesis of UCM, a body of evidence is nowadays accumulating, pointing at micro-RNAs (miRNAs) as new potential key protagonists. miRNAs are short RNA molecules, which are about 20–22 nucleotides in length, transcribed by regions of non-coding DNA, previously known as “junk DNA” [7]. miRNAs target messenger-RNAs (mRNAs), mostly exerting a translational repression and gene silencing. miRNAs are found in all eukaryotic cells, where they play a pivotal role in myriad basic cellular processes, such as cell growth and differentiation [8]. On the other hand, dysfunctional miRNAs activity has been related to many disease conditions, from cancer to autoimmune, neuro-degenerative and inflammatory diseases, as well as to various CV disorders.

Various miRNAs have been implicated in physiological and pathological cardiac remodeling, particularly in the setting of acute and chronic adaptation to ischemic heart disease or in various cardiomyopathies related to heart failure [9,10,11]. However, as said, UCM represents a unique pathological picture where pathogenesis, cellular responses, biomolecular pathways, time evolution and clinical manifestation are of higher complexity and, in some cases, specifically related to ESKD.

In this review, we aimed at summarizing the evidence currently available regarding the involvement of miRNAs in the complex pathogenesis of UCM, their potential usefulness as diagnostic or prognostic biomarkers and their eventual therapeutic application for ameliorating heart damage in the presence of renal disease.

## 2. General Mechanisms Leading to Cardiac Alterations in CKD

Pathological cardiac hypertrophy and fibrosis, eventually determining a progressive decline in the overall cardiac function, is a general, non-specific response that can be found in several diseases, thereby representing a common final pathway that is usually irreversible. As briefly alluded to before, this picture is also a hallmark of UCM. Asymmetric Left Ventricular Hypertrophy is present in about 30% of all patients with CKD and in 70–80% in patients with ESKD. LVH is an independent predictor of mortality in renal patients and may represent the earliest manifestation of cardiac involvement, thereby anticipating the appearance of fibrosis, although in some cases, the two aspects may coexist from the beginning [12].

The main contributing mechanisms to LVH in CKD can be classified into preload-related factors, afterload-related factors and non-afterload-non-preload related factors.

Afterload-related factors are represented by increased systemic arterial resistance and systolic hypertension, which cause a sustained left ventricular overload, leading to maladaptive changes in the cardiomyocytes with ensuing eccentric hypertrophy, left ventricular dilatation and reduced ejection fraction (EF). Preload-related factors mostly rely on an expansion of intravascular volume that eventually ends up, in a similar way to afterload-related factors, in eliciting an eccentric or asymmetrical left ventricular remodeling due to the mechanical stress on the cardiomyocytes. Additionally, non-afterload-non-preload related factors have a plethora of causes that are typical of uremia and that can exacerbate the detrimental effect of preload and afterload factors or contribute independently to cardiac damage. These may include, as a matter of example, toxins accumulation, endothelial dysfunction, oxidative stress and sustained inflammation. Irrespective of the underlying cause(s), myocardial fibrosis, characterized by diffuse collagen deposition between capillaries and cardiomyocytes, leads to a subsequent dilatation of the heart, which indicates a change in the pattern of cellular activity, which pertains not only cardiomyocytes, but also other interstitial cellular populations that actively participate in the pathophysiology of the damage [13]. Although cardiac fibrosis is considered to be irreversible, LVH can be ameliorated to some extent, particularly after kidney transplantation.

## 3. miRNAs in CKD-Induced Pathological Cardiac Remodeling

Various miRNAs may play a significant role in the pathological ventricular remodeling induced by CKD. The miR-30 family is highly expressed in the heart and modulates cardiomyocytes autophagy, apoptosis and oxidative stress, either under normal or pathological conditions [14].

In a recent work, Bao et al. investigated the role of miR-30 in promoting LVH in a model of CKD Sprague-Dawley rats [15]. These rats exhibited lower miR-30 expression, which was inversely related to the entity of renal damage. miR-30 transfection was able to attenuate cardiac apoptosis, thereby suggesting a pivotal role of this miRNA in CKD-induced hypertrophy. Of note, such as inhibition was likely to depend on an inhibition of the calcineurin pathway, a well-known powerful driver of cardiac hypertrophy in various disease models [16].

Several pathogenic factors may inhibit cardiac miR-30 expression, such as oxidative stress, inflammation and hemodynamic disorders [14], the majority of which are also hallmarks of CKD. In this view, miR-30 could thus represent a final common pathway to all these different mechanisms leading to LVH, thereby asking the question as to whether this miRNA could also represent a potential therapeutic target for CKD patients with LVH.

Cardiomyocyte apoptosis plays a crucial role in pathological cardiac remodeling, particularly in the genesis and worsening of asymmetric patterns of LVH, such as eccentric hypertrophy. Similar to apoptosis, pyroptosis is programmed cell death related to inflammation, and it is mediated by caspase-1 activation followed by the cellular processing of proforms of the inflammatory cytokines IL-1β and IL-18 into their active forms [17]. However, compared to other mechanisms of cell death, such as apoptosis and necroptosis, pyroptosis is morphologically and mechanistically distinct. In fact, while apoptosis is induced by caspases 2, 8 and 9 and developed by the effector caspases 3, 6 and 7, pyroptosis is mainly driven by caspase-1 activation. The initial phase of pyroptosis is not distinguishable from apoptosis, while differences occur when the inflammatory process takes over. Macrophage tissue infiltration is commonly found in cardiomyopathies, and macrophage-derived exosomes induce inflammation [18] and promote programmed cell death [19].

miR-155 is an acknowledged promotor of interstitial inflammation and pyroptosis when an extensive cardiac remodeling occurs such as, for instance, after diffusing ischemic heart damage; miR-155 is highly expressed by macrophages that infiltrate the interstitium and can be transferred into cardiomyocytes, thereby regulating their survival [20]. In a landmark study [21], Wang et al. investigated the possible role of miR-155 in regulating cardiomyocyte pyroptosis and hypertrophy in a rat model of CKD with superimposed UCM. Increased levels of caspase-1, IL-1β and IL-18 at the uremic heart level confirmed the implication of pyroptosis in the genesis of UCM. Moreover, a remarkable increase in miR-155 expression was found, particularly in exosomes of infiltrated macrophages. In line with these observations, selective blockades of the caspase-1 pathway attenuated the overall severity of UCM, leading to a dramatic reduction in myocardial hypertrophy and interstitial fibrosis with a following decrease of heart size, heart weight and heart weight-to-body weight ratio and a significant improvement in the Ejection fraction (EF). A similar response was observed in miR-155 knockout models or after administration of selective miR-155 inhibitors while the transfection of macrophage-derived exosomes into uremic miR-155^−/−^ mice restored the UCM picture.

Unlike miR-155, miR-26a was found to be a “beneficial” modulator of cardiac alterations driven by CKD. In another study, Wang et al. [22] demonstrated the capacity of this miRNA to significantly improve peripheral muscle wasting and atrophy, two well-acknowledged consequences of sustained uremia, as well as to reduce fibrosis and hypertrophy at the cardiac level. Despite CKD contributing to UCM and muscle atrophy because of multiple factors, including metabolic acidosis, inflammation and increased oxidative stress, insulin resistance is recognized to play an equally important role. Downregulation of the Insulin/IGF-1 pathway signaling observed in CKD induces muscle wasting by increasing protein degradation through ubiquitin proteasome pathway activation [23]. Interestingly miR-26a expression was found to be almost suppressed on the heart and muscle levels. Transfection of exosomes enriched with miR-26a improved cardiac and muscle sensitivity to insulin by activating Akt and, in turn, by inactivating the FoxO1 and GSK-3 pathways. No less important, miR-26a transfection reduced the circulating levels of the profibrotic factors TGF-β1 and PTEN. Hence, this miRNA may be a candidate as a serious therapeutic target for reverting UCM in CKD.

Forkhead transcription factors of the O class, such as FoxO1, FoxO3, FoxO4 and FoxO6, regulate several genes expression involved in various cell processes, such as proliferation, inflammation, apoptosis and pyroptosis. Activation of FoxO1 and FoxO3a plays a main role in cardiac hypertrophy [24] but unlike FoxO1, FoxO3 levels are notably reduced in uremic hearts. Overexpression of FoxO3a in uremic hearts ameliorates pyroptosis and improves myocardial hypertrophy, interstitial fibrosis area and cardiac function, confirming that FoxO3a decrease is likely to be implied in the pathologic cardiac remodeling occurring in UCM as well [21].

The Mir-212/132 is a miRNA family known to promote pressure-overload-induced LVH and heart failure through the repression of FoxO3 [25]. In a milestone study, Sarkozy et al. [26] found an increased left ventricular expression of miR-212 in rats with CKD as compared to sham. However, FoxO3 expression and phosphorylation levels were unchanged in this model. Bearing in mind that an increased phospho-FoxO3/total-FoxO3 ratio is typical of pressure-overload-induced LVH, the molecular pathways leading to LVH could therefore be different according to the origin of cardiac damage, as well as its mutual interactions. Unexpectedly, this study did not find changes in the levels of other miR-212 targets, such as ERK1, ERK2, Mef2a, AMPK, Sirt1 and PTEN in CKD animals. Hence, other unknown interactions could have played a role in regulating the downstream process of LVH as well as cardiac fibrosis in the course of chronic renal damage.

In another study [27], Chuppa et al. analyzed the expression of miR-21-5p in relationship to the cardiac dysfunction and pathological remodeling occurring in a rat model of CKD induced by 5/6 partial nephrectomy (5/6 Nx). A relevant increase in miR-21-5p levels was found 7 weeks after the establishment of CKD. Additionally, this miRNA exerted clear regulatory activity on the peroxisome proliferator-activated receptor alpha (PPARα). PPARα is involved in inflammation [28] and atherosclerosis [29], and was identified as a translationally repressed target of miR-21-5p [30]. Protein expression of PPARα was significantly reduced with 5/6 Nx surgery only at week 7, the same time point in which miR-21-5p resulted increased, while a selective anticipatory suppression of miR-21-5p expression increased PPARα levels. A low therapeutic dose of the PPARα agonist clofibrate was thus delivered much earlier than the observed increase in miR-21-5p. Although clofibrate was not able to improve kidney function, that drug prevented the decrease in ejection fraction and the development of a significant LV dilation. Of note, no changes were reported on systolic and diastolic blood pressure values, thereby suggesting that such cardiac benefits were independent from a reduction in afterload.

The impact of a panel of cardio-specific miRNAs on the PPARα signaling, as well as on other intracellular pathways, was further investigated in another experimental model of CKD following chronic heart disease (CHD), also known as type-2 cardio-renal syndrome (CRS) [31]. Interestingly, in this study, CHD was induced by pulmonary artery constriction (PAC), an experimental model of pulmonary hypertension that ends with the induction of a pathological, dysfunctional remodeling that affects mostly the right ventricle (RV). Indeed, pulmonary hypertension (PH) is a highly pervasive condition among CKD patients, also representing an independent predictor of adverse CV outcomes [32,33]. miR-205-5p and miR-208b-3p were significantly upregulated in RV specimens from the PAC group as compared with controls, while no differences were reported for miR-21a-3p. Conversely, other miRNAs such as miRNA-215, miRNA-150 and miR-26b-5p were significantly down-regulated. Molecular mechanisms of RV dysfunction were investigated through the construction of pairs of miRNAs to their target genes, in order to get functional information about the biological process implicated. Further analyses of upregulated mRNAs in RV tissue revealed positive regulation of the Ras protein signaling pathway, while downregulated mRNAs showed implications in the low-density lipoprotein receptor particle metabolic process, as well as in carboxylic and organic acid biosynthetic processes. KEGG analyses showed upregulated mRNAs in PAC mice paired with an increased PI-3K/Akt signaling and focal adhesion pathway, while downregulated mRNAs were implicated in the PPARα signaling pathway.

## 4. miRNAs, UCM and the Na/K-ATPase Signaling Pathway

Na/K-ATPase signaling is another major driver of pathological cardiac fibrogenesis. This holds true particularly in UCM, in which a hyperstimulation of this pump is triggered by endogenous steroids, such as marinobufagenin (MBG), which accumulates in the bloodstream as uremic toxins [34,35]. Na/K-ATPase signaling involves downstream signaling proteins, such as Src [36], Akt and PKCs [37], which culminate in the enhancement of various pro-fibrotic pathways [38].

Drummond et al. showed the important relationship between Na/K-ATPase signaling and miR-29b-3p regulation in determining cardiac fibrosis in a rat model of 5/6 partial nephrectomy induced-CKD [39]. The authors found miR-29b-3p expression to be negatively regulated by the Na/K-ATPase signaling through activation of Src and NF-kB. Downregulation of this miRNA was more evident in cardiomyocytes, although it was observed in other interstitial populations as well. An increase in Src and NF-kB activation resulted in fibrosis enhancement in LV tissue samples, thus suggesting an antifibrotic role of miR-29b-3p by acting on the Na/K-ATPase signaling.

In another work [40], the same group confirmed the crucial role of miR-29b-3p in modulating collagen synthesis by cardiac fibroblasts induced by Na/K-ATPase activation [40]. Primary cardiac fibroblasts cultures were obtained from 5/6 Partial Nephrectomy (PNx) and MBG-infused rats. In LV specimens from both PNx and MBG-infused rats, miR-29b-3p expression was decreased respectively by 60% and 50%, while COL1A1 mRNA expression was increased. In vitro, cardiac fibroblasts cultures from adult rats were treated with two different Na/K-ATPase ligands, ouabain and MBG, both eliciting a significant decrease of miR-29b-3p levels and a significant increase in collagen protein expression. Hence, collectively, these data indicate that Na/K-ATPase signaling has a major role in uremia-induced cardiomyopathy through the downregulation of miR-29b-3p expression.

## 5. miRNAs in the RAS-Mediated Enhancement of Fibrosis in UCM

Hyperactivation of the Renin-angiotensin-aldosterone system (RAS) is another worsening factor, as well as a putative driven by cardiac fibrosis in UCM. Angiotensin-II (Ang-II), the last active compound generated in the pathway, exerts its actions through two distinct G-protein-coupled receptors, type 1 (AT1R) and type 2 (AT2R), expressed in different tissues, including the heart. Ang-II has a known effect on collagen synthesis and in the fibronectin stimulating TGF-β1 pathway [41,42]. Classical RAS activation also induces ADAM17, which promotes inflammation and fibrosis through the release of pro-fibrotic and pro-inflammatory cytokines. Previous studies have shown that cardiac fibrosis induced by Ang-II can be prevented by the administration of the small angiotensin-derived peptides Ang (1–7). Moreover, the overexpression of the angiotensin-converting enzyme 2 (ACE2) in mice reverses not only cardiac fibrosis, but also cardiac hypertrophy, while a lower ACE2 expression resulted in the progression of cardiac and renal fibrosis [43].

RAS inhibitors such as ACE-inhibitors (ACEi) and Ang-II receptor blockers (ARBs) are notoriously endowed with anti-fibrotic properties, which make these drugs as best-practice treatment for retarding cardiac and kidney fibrosis in various heart and renal diseases. Evidence nowadays exists that such an antifibrotic effect might be attributable, at least in part, to their capacity to modulate gene transcription of certain miRNAs directly or indirectly involved in cardiac fibrosis and pathological hypertrophy [44].

For instance, in a diabetic nephropathy (DN) mice model, the capacity of the valsartan to ameliorating cardiac fibrosis is partly linked to the inhibition of miR-21 expression [45]. Previous studies evidenced that miR-21 levels in cardiac tissue could be an indirect indicator of cardiac function [46]. Western blotting analysis after transfection experiments demonstrated that miR-21 over-expression significantly decreased the expression of the matrix metalloproteinase-9 (MMP-9), a powerful antifibrotic interstitial protein while, predictably, miR-21 inhibition led to an enhancement of MMP-9 expression. Therefore, miR-21 directly downregulates MMP-9, contributing in this way to cardiac fibrosis in the course of DN. Of note, ARB administration decreased miR-21 expression both in cardiac tissue and in the bloodstream, suggesting that the ARB-mediated suppression of miR-21 expression could be one of the underlying mechanisms of diabetic cardiomyopathy in DN mice. Additionally, this study found the urinary albumin to creatinine ratio (ACR) to be strictly related to miR-21 cardiac levels; this observation paired well with the consolidated assumption that in heart failure with a preserved ejection fraction rate, albuminuria has an independent association with cardiac remodeling [47].

In another study, ACE-inhibition with ramipril was able to prevent cardiac damage in a rat model of cardiac damage following acute kidney injury (AKI) [48]. In these animals, AKI was followed by an increase in miR-212 and miR-132 expression, two miRNAs which are well acknowledged for their pro-fibrotic effect based on the suppression of the antihypertrophic FoxO3 target gene [49]. However, the administration of ramipril blunted such an increase, also eliciting a hyperexpression of miR-133 and miR-1, two acknowledged cardioprotective and anti-fibrotic miRNAs [50]. The target genes of miR-133, such as the profibrotic COL1A1 and the proapoptotic Caspase-9, were upregulated in AKI rats and downregulated in the ACE inhibitor treatment group. By the same token, ramipril administration induced miR-1 upregulation, which caused inhibition of the miR-1 target profibrotic gene FN1 [51] and the antiapoptotic BCL2 gene [52], thereby contributing through another pathway to fibrosis attenuation.

## 6. miRNAs Interplay with Circulating Hormones to Exacerbate UCM

Beyond the RAS, many other hormonal factors may make a significant contribution to UCM worsening. Vitamin D plays an essential role in calcium homeostasis and bone metabolism, but it is also endowed with myriad pleiotropic effects on other tissues and systems [53,54]. Vitamin D receptor activators (VDRAs) have shown anti-inflammatory and antifibrotic effects through the inhibition of specific signaling factors, such as NF-kB, TGF-β1, collagen I and ADAM 17. Hence, activation of Vitamin D-mediated pathways, including Klotho overexpression, prevent worsening of kidney and cardiac damage by inhibiting fibrosis [55]. Importantly, the VDRA paricalcitol attenuates LVH and cardiac fibrosis by reducing ECM proteins synthesis and TGF-β1 expression and by increasing the activity of the MMP-1 collagenase [56,57].

In a rat model of CKD, VDRAs administration was able to attenuate uremia-induced cardiac fibrosis by regulating miR-29b, miR-30c and miR-133b expression [58]. The levels of these miRNAs were reduced in the LV of CKD rats, while calcitriol administration partially prevented this decrease. Mechanistic analyses revealed that such miRNAs specifically targeted the profibrotic genes COL1A1, MMP-2 and CTGF, which suggested a potential therapeutic application as antifibrotic agents.

Like Vitamin-D, thyroid hormones (THs) are also renowned for their pleiotropic effects on several organs including the CV system. A reduction in circulating THs may enhance various mechanisms underlying myocardial damage in the course of uremia [59,60]. The usual hormonal pattern in CKD individuals is represented by low or low–normal plasma levels of triiodothyronine (T3) and thyroxine (T4) with normal thyroid stimulating hormone (TSH). Additionally, peripheral resistance to THs activity may also be present [61]. THs regulate expression of hypertrophy-associated proteins such as the α-myosin heavy chain (α-MHC) and the β-myosin heavy chain (β-MHC) [62], as well as collagen deposition [63] and collagen removal [64]. Abnormal myocardial remodeling may thus occur, driven by a decreased expression of α-MHC and an overexpression of β-MHC.

miR-208, which is selectively expressed in myocardial tissue, modulates α-MHC expression and myocardial fibrosis in response to various stimuli, thereby acting as an important regulator of heart remodeling [65]. miR-208 was previously studied in several models of LVH, such as thoracic aorta banding (TAB), in which this miRNA played a crucial role in shifting the synthesis of α-MHC to β-MHC. Yet, the presence of miR-208 appeared necessary but not sufficient to generate LVH or to modify the MHC pattern, as additional mechanical stimuli are needed to produce such changes [66].

Interestingly, THs may control miR-208 activity at the pre-transcriptional level and the selective antagonism of miR-208 has been seen to be effective in improving cardiac function and survival in heart failure [67]. Accordingly, the ventricular expression of miR-208 was dramatically reduced in a mice model of CKD and UCM while MHC levels were abnormally increased [68]. Of note, the lower miR-208 expression paired well with the severity of LVH and fibrosis and the preventive administration of high doses of T4 was able to normalize miR-208 expression and MHC levels with a subsequent improvement in cardiac geometry and function with, however, no changes reported in kidney damage and function.

Gender-specific hormones are other well-acknowledged protagonists in the progression of renal disease, also exerting a significant impact on long-term CV outcomes [69,70]. Men with CKD usually display a faster tendency to ESKD and a higher incidence of CKD-related CVD; conversely, women show a higher age-adjusted prevalence of CKD and are less exposed to CV disease by the protective effects of gonadal hormones until menopause [71]. The latter is partly dependent on a favorable cardiac morphology and a generally more compliant and preserved CV system.

In a very interesting work, Paterson et al. investigated the sex-specific role of miR-146b-5p on the cardiac pathology in a rat model of CKD [72]. Following 5/6 Nx, cardiac hypertrophy was less pronounced in female miR-146b−/− rats as compared to the wild-type (females) while, on the contrary, renal dysfunction was apparently more pronounced. Such a phenotype was partly attenuated by a preventive ovariectomy. Conversely, the loss of miR-146b-5p had no effect on the hypertrophic response in male animals which, on the contrary, exhibited a significant LVH with chamber dilation following 5/6Nx when compared to miR-146b−/−, but not when compared to wild-type males. Complex in silico analyses evidenced that miR-146b-5p activity mostly targeted the expression of TGFB1 protein in the TGF-β pathway as well as that of Vimentin (Vim) and e-cadherin (Cdh1). Treatment with β-estrogens decreased the TGF-β-induced Vim expression, and pre-miR-146b treatment significantly reduced the expression of collagen in cells simultaneously treated with TGF-β and β-estrogens but not in those incubated with TGF-β alone, thus suggesting a sex hormones-dependent action of miR-146b-5p. Hence, miR-146b-5p in WT females may enhance the protective effects of estrogen, thereby explaining why miR-146b-5p suppression resulted in a dysregulation of gonadal hormones signaling and CKD worsening. Further studies are awaited to improve insights on the possible role of other miRNA families in modulating the effects of sex-hormones on cardiac pathological adaptations in the course of CKD.

## 7. Clinical Studies of miRNAs in CKD/ESKD Individuals

Despite such a large body of experimental research accrued on the involvement of various miRNAs in the genesis and worsening of UCM (Table 1 and Figure 1), the number of clinical studies on miRNAs measurement in renal patients in relationship to cardiac alterations or CV outcomes is, so far, rather disappointing.

Wen et al. analyzed circulating miR-133a as a biomarker of LVH in 64 chronic hemodialysis (HD) patients [73]. The plasma level of miR-133a was lower in 40 HD patients, displaying evidence of LVH as compared to those without. Additionally, circulating miR-133a levels were lower with respect to controls and negatively associated with the left ventricular mass index (LVMI). Of note, miR-133a levels in the blood remained unchanged after a single HD session, despite the molecular weight of miRNAs being small enough to permeate the dialysis membrane.

miR-133a was also analyzed in another prospective cohort study of 87 patients with CKD of various degrees [74]. Despite miR-133a levels being altered in these patients, no correlations were found with the entity of LVH or with the occurrence of a composite endpoint of CV death or CV events over an 18-month follow-up.

Recently, according to [75], our group investigated a small, circulating miRNAs panel in a multicentric cohort of 74 HD patients who were prospectively followed considering an established composite endpoint of mortality and CV events. Circulating levels of miRNAs 23a-3p, 451a, 30a-5p and let7d-5p were all lower in chronic HD patients as compared with healthy individuals and significant correlations were found between miRNAs and the indexes of cardiac dysfunction such as Vmax, TAPSE and E/E’. All miRNAs but let7d-5p were more reduced at the baseline in HD patients experiencing the composite endpoint and were revealed as significant, and as predictors of adverse outcomes in either unadjusted or multivariate survival analyses. Hence, measurement of a selected, circulating miRNAs panel may hold usefulness for improving CV risk stratification in a very high-risk setting represented by ESKD. Evidence from clinical studies on miRNAs in UCM are summarized in Table 2.

## 8. Overview and Future Directions for Studies on miRNAs in UCM

As widely described for many known disease conditions, various miRNAs could play a major role in the pathogenesis of UCM, the most important and pervasive CV complication found among renal patients. Nevertheless, the exact biological significance and mechanistic contribution to the origin, development and worsening of UCM remains, for the majority of miRNAs are still far from being completely understood. This is, at least in part, the consequence of the extreme complexity of this pathological condition that recognizes a plethora of uremic and non-uremic predisposing factors, some of which remains unknown.

Future advancement in the comprehension of the role of miRNAs in UCM could yield important benefits for clinical practice. First, it may shed further light on understanding the crosstalk between different molecular and signaling pathways, thereby getting new insights in the pathophysiology of this condition which would facilitate the identification of new causative factors and, above all, new potential therapeutic targets. In this latter regard, some experimental studies have already tested the potential usefulness of some miRNAs as possible therapies for ameliorating the individual components of UCM, namely fibrosis and LVH. Unfortunately, such evidence remains, to date, confined to few animal models but the extremely promising findings obtained would ideally set the stage for future testing of miRNAs therapy in the human setting as well.

Last, but not least, the measurement of specific circulating miRNAs holds the potential utility of improving and easing the diagnosis and the severity staging of UCM, as well as the long-term prognosis, in a non-invasive and reproducible way. Forthcoming clinical research in this latter regard is eagerly awaited to identify, more likely than a single molecule, a multimolecular panel of miRNAs putatively involved in UCM pathogenesis to evaluate the impact of their measurement in improving CV risk prediction, in addition to the traditional factors and the individual predisposition.

## Figures and Tables

**Figure 1 ijms-24-05425-f001:**
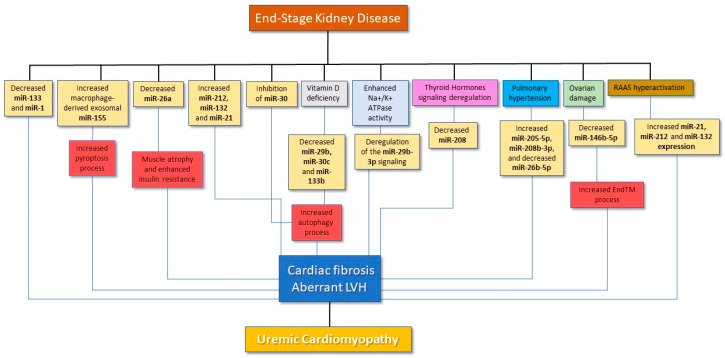
Main pathways linking miRNAs to the development and worsening of UCM.

**Table 1 ijms-24-05425-t001:** Experimental Studies of miRNAs in UCM.

Study	Methods and Model	miRNAs	Key-Findings
- Bao et al. [15]	- 5/6th partial nephrectomy mice - Sham-operated mice	- miR-30	- miR-30 is highly expressed in normal hearts and downregulated in the CKD model - miR-30 attenuated the development of cardiac hypertrophy in CKD without influencing CKD progression - miR-30 suppression is related to calcineurin activation in CKD-induced LVH. - FGF-23, uremic toxins, ANG2, and TGF–β suppressed cardiac miRNA-30 expression - miRNA-30 supplementation blunted cardiomyocyte hypertrophy
- Wang et al. [21]	- miR-155−/− mice - Wild-type male mice - 5/6th partial nephrectomy mice	- miR-155	- Caspase-1, IL-18 and IL-1a (IL-1β?) levels were increased in CKD rats - FoxO3a was significantly decreased in uremic hearts as compared with the control group - Inhibition of miR-155 expression attenuated pyroptosis and UCM by targeting FoxO3a 3′-UTR - Macrophage-derived exosomal miR-155 is critical in cardiomyocyte pyroptosis in uremic hearts through downregulation of FoxO3a protein expression - The miR-155/FoxO3a axis induced pyroptosis in UCM - Infiltrated macrophages secreted miR-155 enriched exosomes and promoted cardiomyocyte pyroptosis by directly targeting FoxO3a in UCM
- Wang et al. [22]	- 5/6th partial nephrectomy mice - Sham-operated mice - CKD Exo/miR-26a mice - CKD Exo/ctrl mice	- miR-26a	- miR-26a-5p attenuated CKD-induced muscle atrophy - miR-26a was decreased in the heart and skeletal muscle of CKD mice - Injection of Exo/miR-26a reduced uremic cardiomyopathy in CKD mice - miR-26a limited insulin resistance with improvement of cardiac function - FoxO1 is a direct target of miR-26a. - Muscle atrophy and uremic cardiomyopathy are related to a decrease in miR-26a
- Sarkozy et al. [26]	- 5/6th partial nephrectomy mice - Sham-operated mice	- miR-212	- LV expression of miR-212 significantly increased in CKD mice as compared to the sham group - phospho-Akt/total Akt ratio significantly increased in CKD - phospho-mTOR/mTOR ratio did not change after CKD induction - Molecular markers of pro-hypertrophic calcineurin/NFAT pathway did not change in CKD as compared to the sham group
- Chuppa et al. [27]	- 5/6th partial nephrectomy mice (PNx) - Sham-operated mice	- miR-21-5p	- PPARα is regulated by miR-21-5p and its activation improved renal and cardiac function - miR-21-5p suppression in 5/6PNx mice is cardioprotective - Chronic knockdown of miR-21-5p prevented hypertrophic remodeling of the LV wall in 5/6PNx mice - anti–miR-21-5p knockdown prevented subsequent insulin-like growth factor-1–induced increases in cultured cell area - Clofibrate treatment (PPARα-agonist) increased systolic wall thickness and significantly reduced LV dilation.
- Chen et al. [31]	- RV dysfunction-induced type II CRS mouse model by pulmonary artery constriction (PAC) - Sham-operated mice	- miR-21a-3p - miR-205-5p - miR-208-3p - miR-215-5p - miR-26b-5p - miR-150	- miR-205-5p and miR-208b-3p were significantly upregulated in the RV tissue in the PAC group as compared with the sham group - No significant differences in miR-21a-3p between the two groups - miR-215, miR-150 and miR-26b-5p were involved in fibrotic- and proliferative-related pathways by targeting FoxO4 and Kremen1
- Drummond et al. [39]	- 5/6th partial nephrectomy mice (PNx) - Na/K-ATPase a1 heterozygous (α1+/−) mice (40% less Na/K-ATPase a)	- miR-29b-3p	- Na/K-ATPase signaling regulates miR-29b-3p through activation of Src and NF-kB - Trend of miR-29b-3p was the opposite of that of Src and NF-kB activation - NF-kB is involved in the Na/K-ATPase-mediated miR-29b-3p regulation - Na/K-ATPase signaling downregulates miR-29b-3p through Src activation by phosphorylation of Tyr418
- Drummond et al. [40]	- 5/6th partial nephrectomy mice - MBG-infusion	- miR-29b-3p	- CTS induced Na/K-ATPase signaling mediates miR-29b-3p expression in cardiac fibroblast - miR-29b-3p levels regulated by activation of proteinkinase C δ (PCKδ), and degradation of friend leukemia integration 1 (Fli-1), while no increases in TGF-β/Smad signaling were observed, despite inhibition of TGF-β blocked collagen synthesis inducted by MBG - downregulation of miR-29b-3p increased cardiac fibrosis
- Wang et al. [45]	- KK-Ay mice (DN model) treated with vehicle or ARB (valsartan) - KK-Ay DN mice treated with ARB + pre-miR-21 lentivirus vector injection	- miR-21	- miR-21 expression is markedly increased in DN - ARB decreased miR-21 expression in cardiac tissue and in serum - miR-21 overexpression increased col-IV and FN mRNA and protein expression - ARB blunted col-IV expression by inhibiting miR-21 expression - miR-21 overexpression significantly reduced MMP-9 expression - miR-21 expression was significantly decreased in the LV tissue in the DN and premiR-21 group
- Rana et al. [48]	- CKD rats (nephrectomy of the right kidney and selective ligation of the left kidney) - Sham-operated mice - ACEi treatment with ramipril	- miR-1 - miR-133 - miR-212 - miR-132	- Cardiac expression of miR-212 and miR-132 (prohypertrophic) is upregulated in CKD rats and suppressed antihypertrophic FoxO3 - ACEi treatment attenuated miR-212/132 expression and increased miR-133/1 expression - Profibrotic Col1A1 and apoptotic caspase-9 are upregulated in CKD rats and downregulated after treatment with ACEi - Increased miR-1 inhibited the profibrotic gene Fn1 and upregulated the antiapoptotic gene BCL-2 - Increased miR-133 decreased caspase-9 mRNA and inhibited mitochondrial apoptotic pathway
- Panizo et al. [58]	- 7/8th partial nephrectomy mice - 4-week treatment with intraperitoneal infusion of the two vitamin D receptor activators (VDRAs) calcitriol and paricalcitol	- miR-29b - miR-30c - miR133b	- Both VDRAs reduced fibrosis, which was more evident after paricalcitol treatment - Cardiac expression of COL1A1 and MMP-2 increased significantly in CKD mice receiving vehicle - Mice receiving calcitriol did not exhibit a significant change in either cardiac COL1A1 or MMP-2 levels as compared with CKD mice - Cardiac levels of miR-29b, miR-30c and miR-133b were modified by uremia and VDRAs treatment
- Prado-Uribe et al. [68]	- 5/6th partial nephrectomy mice (PNx) - Sham-operated mice - T4 administration	- miR-208	- miR-208 levels in LV wall decreased in 5/6PNx mice but not in 5/6Nx mice after T4 administration - MHC levels in LV wall increased in 5/6PNx mice but not in 5/6PNx mice after T4 administration
- Paterson et al. [72]	- miR-146b-5p −/− mice - WT mice - 5/6th partial nephrectomy mice (PNx) - Sham-operated mice	- miR-146b-5p	- WT males and females exhibited renal and cardiac hypertrophy after 5/6Nx - miR-146b−/− females tended to develop more renal hypertrophy and less cardiac hypertrophy after 5/6PNx as compared to WT. - miR-146b−/− males exhibited a significant increase in LVH with chamber dilation following 5/6PNx as compared to miR-146b−/− sham, but not as compared to WT - Increased renal dysfunction was observed in miR-146b−/− females - Ovariectomy in females attenuated renal pathology and abolished genotypic differences after 5/6PNx.

**Legend:** ACEi: Angiotensin-Converting Enzyme inhibitors; ANG2: Angiotensin II; Akt: protein kinase B; ARB: Angiotensin Receptor Blockers; BCL-2: B-cell lymphoma 2; CKD: Chronic Kidney Disease; COL1A1: collagen type I alpha 1 chain gene; Col1-a1: collagen type I alpha 1 chain protein; Col-IV: collagen type IV; CTS: cardiotonic steroids; DN: diabetic nephropathy; Exo: exosome; FGF-23: Fibroblast Growth Factor 23; FN: fibronectin; Fn1: fibronectin 1 protein; FOxO1: Forkhead Box O1; FOxO3: Forkhead Box O3; FOxO4: Forkhead Box O4; Kremen1: Kringle-Containing Transmembrane Protein 1; IL-1a: interleukin 1 alpha; IL-1β: interleukin-1 beta; IL-18: interleukin-18; LV: left ventricle; LVH: left ventricular hypertrophy; MBG: marinobufagenin; MMP-2: matrix metalloproteinase 2; MMP-9: matrix metalloproteinase 9; mTOR: mammalian target of rapamycin; NFAT: Nuclear Factor of Activated T-Cells; NF-kB: nuclear factor kappa B; PKC: protein kinase C; PPARα: Peroxisome Proliferator Activated Receptor Alpha; RV: right ventricle; Smad: small mothers against decapentaplegic family transcription factors; TGF-β: Transforming Growth Factor beta; T4: thyroxine; UCM: uremic cardiomyopathy; VDRAs: vitamin D receptor activators; WT: wild type.

**Table 2 ijms-24-05425-t002:** Clinical Studies of miRNAs in UCM.

Study	Design	Population/Methods	miRNAs	Results
- Wen et al. [73]	- Cross-sectional	- 64 ESKD patients on chronic HD (40 with LVH)	- miR-133a	- Plasma levels of miR-133a were lower in HD patients as compared with controls and more decreased in HD patients with LVH as compared with those without - Plasma miR-133a levels were inversely associated with LVMI - miR-133a levels were unchanged after a single HD session
- Stopic et al. [74]	- Prospective	- 87 subjects with CKD of various degrees - 18-month follow-up - Endpoints: CV events/ CV death	- miR-133a	- No association between miR-133a and LVH - No predictive value of miR-133a with respect to the endpoints
- Bolignano et al. [75]	- Prospective	- 74 ESKD patients on chronic HD - 24-month follow-up - Endpoint: All-cause and CV death, CV events	- miR-23a-3p - miR-451a - miR-30a-5p - miR-let7d-5p	- All miRNAs levels reduced in HD patients as compared with healthy controls - miRNAs were correlated with Vmax, TAPSE and E/E’ - Lower levels of all miRNAs but miR-let7d-5p were independently associated with the combined endpoint

**Legend:** CKD: Chronic Kidney Disease; CV: cardiovascular; E/E’: early diastolic mitral annular tissue velocity; ESKD: End-Stage Kidney Disease; HD: hemodialysis; LVH: left ventricular hypertrophy; LVMI: left ventricular mass index; TAPSE: Tricuspid Anulus Plane Systolic Excursion; Vmax: peak aortic valve velocity.

## Data Availability

Not applicable.

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
