# Peer review of "miRNAs in Uremic Cardiomyopathy: A Comprehensive Review"

_ijms, 2023, doi:10.3390/ijms24065425_

Round 1

Reviewer 1 Report

The authors provide a review of published literature related to miRNAs and uremic cardiomyopathy.  This is a timely review of a developing research area.  As such, the manuscript should be of interest to readers in this area.  In general, the manuscript is well organized.  Several suggestions include:

- The manuscript needs careful grammatical editing.  In several cases, it was difficult to understand what the authors intended.

- Suggest adding further details regarding the progression and cellular mechanisms of uremic cardiomyopathy (either around line 52 or 77).

- Similarly a little more detail about what is known regarding the progression of chronic kidney disease and the role(s) of miRs in this.  This may be a good lead-in to miRs in cardiac consequences of chronic kidney disease.

- Table 1 is a bit cumbersome and should be a more concise summary of miRs in uremic cardiomyopathy.

- A graphic figure highlighting miRs in uremic cardiomyopathy to supplement Table 1 might be helpful.   Some of the mechanisms could be included there instead of Table 1. 

Author Response

REVIEWER 1

The authors provide a review of published literature related to miRNAs and uremic cardiomyopathy.  This is a timely review of a developing research area.  As such, the manuscript should be of interest to readers in this area.  In general, the manuscript is well organized.  Several suggestions include:

- The manuscript needs careful grammatical editing.  In several cases, it was difficult to understand what the authors intended.

The manuscript has now been revised by an English native speaker with minor mistakes and misspellings identified and corrected.

- Suggest adding further details regarding the progression and cellular mechanisms of uremic cardiomyopathy (either around line 52 or 77).

-We thank the reviewer for this input. Further details on UCM onset and progression have now been provided, as suggested, from line 55 onwards

- Similarly a little more detail about what is known regarding the progression of chronic kidney disease and the role(s) of miRs in this.  This may be a good lead-in to miRs in cardiac consequences of chronic kidney disease.

-We thank the reviewer for this suggestion. Actually, we think that providing a description of mechanisms of CKD progression and, even more, the involvement of miRNAs in triggering such condition would fall outside the broad scope of this review, thereby hampering the overall readability of our paper. Actually, many other reviews (even systematic) have addressed this issue as a plethora of evidence has been produced investigating the effects of certain miRNAs on CKD progression. In our opinion, this topic would have too little or no overlap with the scope of the present manuscript.

- Table 1 is a bit cumbersome and should be a more concise summary of miRs in uremic cardiomyopathy.

-We have now removed the last column of the table, making it more readable and concise. Information on cellular pathways of miRNAs in UCM have now been summarized in Figure 1 (see next point).

- A graphic figure highlighting miRs in uremic cardiomyopathy to supplement Table 1 might be helpful.   Some of the mechanisms could be included there instead of Table 1.

-A Figure (Fig.1) complementary to Table 1 has now been elaborated summarizing the key pathways linking miRNAs to UCM

Reviewer 2 Report

miRNAs in Uremic Cardiomyopathy: a comprehensive review

Manuscript entitled “miRNAs in Uremic Cardiomyopathy: a comprehensive review” by Mario et al., Is a useful review article. In this current review, the author described the biological and clinical significance of micro-RNAs (miRNAs) in Uremic Cardiomyopathy including the role of miRNAs in various signaling pathways like Na/K-ATPase signaling pathway, RAS-mediated enhancement of fibrosis as well as CKD/ESKD individuals.

Overall, the information presented in this review article is useful to the researchers and I approve its publication after some minor updates. 

Minor comments: I suggest that these comments be updated before publication.

The author has given the table with the experimental studies of miRNAs in UCM but it will be better to have a diagram/flowchart with various miRNAs and their associated molecular pathways in Uremic Cardiomyopathy.

Author Response

REVIEWER 2

Manuscript entitled “miRNAs in Uremic Cardiomyopathy: a comprehensive review” by Mario et al., Is a useful review article. In this current review, the author described the biological and clinical significance of micro-RNAs (miRNAs) in Uremic Cardiomyopathy including the role of miRNAs in various signaling pathways like Na/K-ATPase signaling pathway, RAS-mediated enhancement of fibrosis as well as CKD/ESKD individuals.

Overall, the information presented in this review article is useful to the researchers and I approve its publication after some minor updates.

Minor comments: I suggest that these comments be updated before publication.

The author has given the table with the experimental studies of miRNAs in UCM but it will be better to have a diagram/flowchart with various miRNAs and their associated molecular pathways in Uremic Cardiomyopathy.

-A Figure (Fig.1) complementary to Table 1 has now been elaborated summarizing the key pathways linking miRNAs to UCM